# Using Logical Specifications of Objectives in Multi-Objective Reinforcement Learning

## Abstract

In the multi-objective reinforcement learning (MORL) paradigm, the relative importance of each environment objective is often unknown prior to training, so agents must learn to specialize their behavior to optimize different combinations of environment objectives that are specified post-training. These are typically linear combinations, so the agent is effectively parameterized by a weight vector that describes how to balance competing environment objectives. However, many real world behaviors require non-linear combinations of objectives. Additionally, the conversion between desired behavior and weightings is often unclear.

In this work, we explore the use of a language based on propositional logic with quantitative semantics–in place of weight vectors–for specifying non-linear behaviors in an interpretable way. We use a recurrent encoder to encode logical combinations of objectives, and train a MORL agent to generalize over these encodings. We test our agent in several grid worlds with various objectives and show that our agent can generalize to many never-before-seen specifications with performance comparable to single policy baseline agents. We also demonstrate our agent's ability to generate meaningful policies when presented with novel specifications and quickly specialize to novel specifications.

## 1 Introduction

Reinforcement Learning (RL) is a method for learning behavior policies by maximizing expected reward through interactions with an environment. RL has grown in popularity as RL agents have excelled at increasingly complex tasks, including board games (Silver et al., 2016), video games (Mnih et al., 2015), robotic control (Haarnoja et al., 2018), and other high dimensional, complex tasks. RL continues to be a valued area of research as algorithms become more generalizable and sample efficient, making them more feasible for deployment in real world scenarios.

Many RL tasks can be imagined in which multiple possibly conflicting objectives exist. The relative importance of each objective may not be known by the system designer prior to training, and–when it comes to real world deployment–it may be difficult or impossible to retrain agents as the priorities of objectives change over time. Rather than retrain agents for each prioritization, multiple objective RL (MORL) (Roijers et al., 2013) seeks to learn a set of potential policies so that importance of objectives can be specified after training, thus creating more flexible, adaptable agents.

For example, a cleaning agent in a house environment may have several objectives such as dusting, sweeping, and vacuuming. The agent can learn to complete each of these objectives, but certain objectives may take priority over others. A user may specify that dusting is twice as important as sweeping and that vacuuming is not important at all. These priorities may change, and MORL allows agents to learn policies that can satisfy any prioritization of objectives.

As part of MORL, a *scalarization* function is chosen to convert a multiple objective reward vector into a scalar. The most common scalarization function for MORL is a linear combination in the form of a weight vector. However, many real world scalarization functions are non-linear. Additionally, weight vectors are not ideal for specifying desired behavior. A user may need to experiment in order to find the weights that result in a desired behavior. Thus scalarization functions that are interpretable and allow for non-linearities are preferred.

Returning to the cleaning agent example, a user may specify that the cleaning agent should either sweep or vacuum, but it is not necessary to do both. Perhaps the decision should be determined by whichever is more likely to be done well in consideration of other prioritized objectives. Or, it may be useful to specify that the house should always be kept 75% dusted and that the rest of the agent's time should be spent vacuuming. These specifications become difficult or impossible to encode with simple linear weightings.

Multi-task RL (MTRL) (Caruana, 1997) is a generalization of MORL in which an agent learns to complete multiple tasks simultaneously, often with the ultimate goal of completing some more complex task. Recently, Universal Value Function Approximators (UVFA) (Schaul et al., 2015) were developed for MTRL to learn state or q-values over multiple goals. UVFAs require a goal parameterization as input usually defined as an element of the state (Andrychowicz et al., 2017). However, the environment state is not always expressive enough to parameterize all of the goals we may want to learn. Multiple objectives may provide a better way to define additional goals that are not part of the environment state.

The contributions of this work can be defined in two parts. First, we propose a simple language based on propositional logic to specify logical combinations of multiple objectives. This language is equipped with quantitative semantics that are used to define scalarization functions for MORL. The resulting scalarization functions can express non-linear combinations of objectives and are more interpretable than traditional weight vectors. The language also acts as a way of specifying goals for multi-task learning that will be parameterized to use UVFAs. Second, we develop a MORL agent for use with this language. The agent prepends a recurrent encoder onto a UVFA architecture to parameterize goals specified in our language. The agent also follows a learning curriculum over specifications to improve training speed and performance.

We demonstrate that our agent generalizes to never-before-seen specifications defined in our language by providing a test set of novel specifications; this can be seen as a form of zero-shot RL. Performance on test sets is compared to baseline agents trained on single specifications, and we show that our agent performs comparably to baselines despite having never been trained on the test specifications. We demonstrate this over multiple grid worlds with various objectives. Finally, we demonstrate our agent's ability to quickly specialize to novel specifications post-training.

## 2 BACKGROUND

### 2.1 REINFORCEMENT LEARNING

Traditional RL problems are often modeled by a Markov Decision Process (MDP) defined by the tuple $(\mathcal{S}, \mathcal{S}_0, \mathcal{A}, p)$ where $\mathcal{S}$ is the set of states, $\mathcal{S}_0$ is the set of initial states, $\mathcal{A}$ is the set of actions, and $p(r, s'|s, a)$ defines transition probabilities for the environment. Here $s' \in \mathcal{S}$ is the subsequent state and $r \in \mathbb{R}$ is the reward. An agent's objective is to maximize expected return:

$$U = \mathbb{E}_{s \sim \mathcal{S}} \left[ \mathcal{R}_\pi(s) \right] \tag{1}$$

$$\mathcal{R}_\pi(s) = \mathbb{E}_{r,s' \sim p} \left[ r + \gamma \mathcal{R}_\pi(s') \right] \tag{2}$$

where $\mathcal{R}_\pi$ is the return under a policy $\pi : \mathcal{S} \rightarrow \mathcal{A}$ mapping states to actions, and $\gamma$ is a discount factor.

Q-learning is an approach to solving reinforcement learning problems. The Deep Q-Learning (DQN) algorithm (Mnih et al., 2015) utilizes a neural network $Q$ to estimate the Q-value of a state-action pair. The return under a policy $\mathcal{R}_\pi$ can be expressed by the Q-value when the policy is defined by the network $Q$:

$$\mathcal{R}_\pi(s) = \max_{a \sim \mathcal{A}} Q(s, a) \tag{3}$$

$$\pi(s) = \arg \max_{a \sim \mathcal{A}} Q(s, a). \tag{4}$$

We utilize the DQN algorithm in this paper to build our MORL agent, although our language and the way we construct our agent are not limited to the DQN algorithm.

## 2.2 MULTI-OBJECTIVE REINFORCEMENT LEARNING

We alter the traditional RL formulation as an MDP by using a vector of rewards rather than a scalar reward, resulting in a Multi-Objective Markov Decision Process (MOMDP) (Roijers et al., 2013). We represent a MOMDP as a tuple $(\mathcal{S}, \mathcal{S}_0, \mathcal{A}, p)$ where $p(\mathbf{r}, s'|s, a)$ maps to the reward vector $\mathbf{r} \in \mathbb{R}^n$ rather than a scalar reward. Each of the $n$ elements in $\mathbf{r}$ represents the reward for a particular objective.

Next, we modify the DQN algorithm for MORL. Equation 2 requires a scalar reward to sum with the discounted future rewards. Since the new transition function $p$ provides a vector reward $\mathbf{r}$ we need a scalarization function. Typically, the scalarization function is defined as a linear combination of objectives in the form of a weight vector (Abels et al., 2018). In this work, we explore the use of non-linear scalarizations functions defined by logical combinations of objectives.

In offline MORL, the scalarization function is unknown prior to training, so an agent's goal is to learn a coverage set (Roijers et al., 2013) of policies in which there is at least one optimal policy for any scalarization function. Previously this has been done by finding a convex coverage set of policies (Mossalam et al., 2016). However, this method does not work for non-linear scalarization functions.

Abels et al. (2018) explored the use of UVFAs to train MORL agents in an online setting. In an online MORL setting, an agent must learn on the fly and is not given the opportunity to learn a coverage set prior to testing. Instead the agent must adapt quickly to new scalarization functions without forgetting policies for previously seen scalarization functions. Online MORL is outside of the scope of this paper, but we also utilize UVFAs to instead learn a dense set of policies, trained offline, to be used post training with arbitrary behavior specifications.

## 2.3 UNIVERSAL VALUE FUNCTION APPROXIMATORS

Universal Value Function Approximators (UVFA) (Schaul et al., 2015) learn value functions over multiple goals, taking as input to a neural network a goal parameterization along with environment state or a state action pair. This dense set of value functions induces an equally dense set of policies which can be applied to their respective goals. Empirically, UFVAs are shown to be able to generalize to novel goals. We utilize this method to parameterize goals specified by logical expressions in our language and show that these statements can be generalized across as well. Many similar techniques are applied in multi-task learning scenarios. Although, the purpose of multi-task learning is often to improve the learning speed of an agent rather than improve its ability to generalize to new tasks. The latter is our objective.

## 3 MULTI-OBJECTIVE RL WITH LOGICAL COMBINATIONS OF OBJECTIVES

Rather than learn a set of individual policies to approximate a coverage set over scalarization functions, we train a single agent to generalize over encoded representations of scalarization functions. We define scalarization functions in the semantics of a custom language. Behavior specifications are represented as logic strings, passed through an encoder, and then given to a Q-network for learning. In this section we lay out our language's syntax and semantics, the agent architecture, the encoding method, and the learning curriculum for our agent.

### 3.1 LANGUAGE SYNTAX AND SEMANTICS

We define a language based on propositional logic to specify combinations of objectives. The logic operates over objective rewards between $[0, 1]$ inclusive, and is defined as follows.

### 3.1.1 SYNTAX

We define the grammar of our specification logic as follows:

$$\psi := \psi \wedge \psi \mid \psi \vee \psi \mid o_n \mid \neg o_n \mid o_n \geq c \mid o_n \leq c$$

Here, $o_n$ is the value of the nth objective, or the nth element in a reward vector, and $c$ is a constant. For the purposes of training, we use the current environment's reward function to define the possible

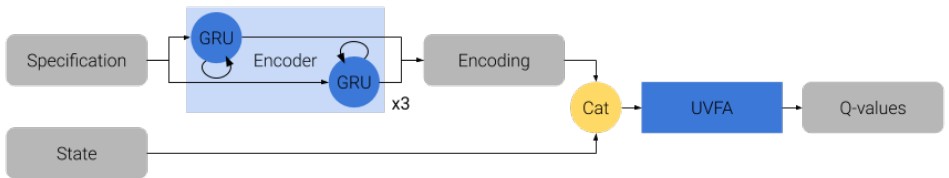

Figure 1: Network architecture: A behavior specification is encoded using a three layer bidirectional GRU, concatenated with the state representation, and used as input to a UVFA to estimate Q-values.

values of c for each objective. Also notice that the $\wedge$ and $\vee$ operations can be performed non-terminal elements of the grammar.

### 3.1.2 SEMANTICS

The quantitative semantics of our specification language are defined below:

$$f(\mathbf{r}, o_n) = \mathbf{r}[n]$$
$$f(\mathbf{r}, \neg o_n) = 1 - \mathbf{r}[n]$$
$$f(\mathbf{r}, o_n \geq c) = 1 \; if \; \mathbf{r}[n] \geq c \; else \; 0$$
$$f(\mathbf{r}, o_n \leq c) = 1 \; if \; \mathbf{r}[n] \leq c \; else \; 0$$
$$f(\mathbf{r}, \psi_1 \wedge \psi_2) = min(\psi_1, \psi_2)$$
$$f(\mathbf{r}, \psi_1 \vee \psi_2) = max(\psi_1, \psi_2)$$

Notice that any objective may be either minimized or maximized according to a soft ($o_n$, $\neg o_n$) or hard ($o_n \geq c$, $o_n \leq c$) constraint. The soft constraints return the value of the objective reward while the hard constraints return a value of one or zero.

With these semantics, we are able to specify complex desired behaviors. Consider once again a cleaning agent tasked with several cleaning objectives: 1) dust, 2) sweep, and 3) vacuum. In this scenario, the reward vector returned by the environment represents what percentage of the house is cleaned sufficiently for each of the objectives. That means our language allows us to instruct an agent to sweep as much as possible, maintain a certain percentage swept, or even make as big of a mess as possible. Additionally, if we instruct an agent to dust **or** sweep, we can expect it to complete one objective and not the other. We can also expect that the agent will choose to complete the objective that can be satisfied more quickly because it will have a higher expected return. With the above defined predicate logic we can express desired behaviors such as:

| | |
|---|---|
| $o1 \geq .5 \wedge o2$ | Dust 50% of the house and sweep as much as possible. |
| $o1 \geq .8 \wedge o2 \geq .8 \wedge o3 \geq .8$ | Keep the house 80% dusted, swept, and vacuumed. |
| $(o1 \wedge o2) \vee (o2 \wedge o3)$ | Either dust and sweep or sweep and vacuum. |
| $o2 \wedge (o1 \vee o3)$ | Sweep and either dust or vacuum. |
| $o1 \wedge \neg o2$ | Dust while increasing the amount of sweeping to be done. |

### 3.2 MULTI-OBJECTIVE DQN

For our MORL agent we implement a vanilla DQN as described by Mnih et al. (2015), although our method is easily applicable to most RL algorithms. Recent advances in MTRL, such as the use of UVFAs for generalizing across goals, has become more common in multiple objective settings (Friedman & Fontaine, 2018; Abels et al., 2018). We also utilize UVFAs to generalize across encoded behavior specifications and output Q-values for each environment action. The system architecture for the MORL agent is diagrammed in Figure 1. Output from the encoder is concatenated with a state representation and passed through a four-layer neural network of hidden size 128 that outputs Q-values for each action in the finite action space.

As the agent gains experience, tuples of state, action, next state, terminal, and reward vector $(s, a, s', t, \mathbf{r})$ are stored in a replay buffer for future training. In our implementation, every 5 steps

a batch of size 32 is sampled from the replay buffer, and every sampled tuple is augmented with 8 different behavior specifications $\psi$. Experience tuples are augmented using the reward vector from the tuple and the semantics of a sampled specification to generate a scalar reward. The loss is then calculated over this batch using the formula:

$$Loss = Q(s, a) - (f(\mathbf{r}, \psi) + \gamma \max_{a' \sim \mathcal{A}} \hat{Q}(s', a')(1 - t)) \tag{5}$$

Here $f : \mathbf{r} \times \psi \to \mathbb{R}$ are the language semantics that map reward vectors and specifications to scalar rewards. Also, $\hat{Q}$ is the target Q-network, part of the typical DQN algorithm, a bootstrapped estimate of the true Q-value updated regularly with the parameters of $Q$.

## 3.3 Language Encoder

We utilize Gated Recurrent Units (GRU) (Cho et al., 2014) in order to encode behavior specifications in the previously defined language. We implement this encoder with three bidirectional layers of hidden size 64 as shown in Figure 1. The input to our encoder is a sequence of one-hot encodings generated from a tokenized specification. The hidden states of the last layer in each direction are used as the specification encoding. The output of this encoder then serves as input to our MORL agent.

We train this language encoder end-to-end along with the rest of our agent. During testing, we found that allowing gradients to flow from the DQN back through the encoder gave us better results than other attempts to train the encoder in a separate supervised setting. In the supervised training scenario, we trained using the output of the encoder along with a sampled reward vector to predict the scalarized reward produced by the specification's semantics (the idea being that the encoded specification should retain sufficient information to predict the scalarized reward). However, after several experiments with supervised pretrained and co-trained encoders, we determined that end-to-end training provided the DQN agent with better specification encodings.

## 3.4 Learning Curriculum

We started by training our agent with a new, randomly sampled behavior specification for each training episode. Additionally, we randomly sampled behavior specifications to augment training batches. However, we wondered if it would struggle to generalize to long, complex specifications.

We therefore experimented with using curricula to slowly increase the complexity of sampled specifications. Under this curriculum, we only use a subset of all behavior specifications when sampling for environment episodes and batch augmentation. This subset of behaviors is defined by the maximum length of specification strings in the subset. In our final agent, we increment the maximum length of specifications in the subset every 5000 timesteps. The curriculum increments the maximum length a total of 20 times, starting at a base length of 25 characters up to the maximum specification length for the entire set of specifications.

We compare learners trained with and without a learning curriculum in the experiments in Section 4.

## 4 Experimental Results

We present the following experiments and results to demonstrate our agent's ability to encode the quantitative semantics of behavior specifications and behave appropriately according to never-before-seen specifications.

The gridworld shown in Figure 2 is a diagram of the environment used to test our MORL agent's ability to generalize to new behavior specifications in a multi-objective setting. We refer to this as the navigation environment. It has three objectives: 1) stay on the road, 2) avoid hazards, and 3) move right. The first is marked by increasingly dark cells indicating increasing reward. Reward for the second objective decreases starting inside the red outline until it reaches zero at the location of the darkest cell. Finally, the third objective reward increases closer to the right side of the grid.

Environment state is finite and can be represented by the location of an agent in the grid. The environment's action space is also finite and composed of up, down, left, and right movements. All

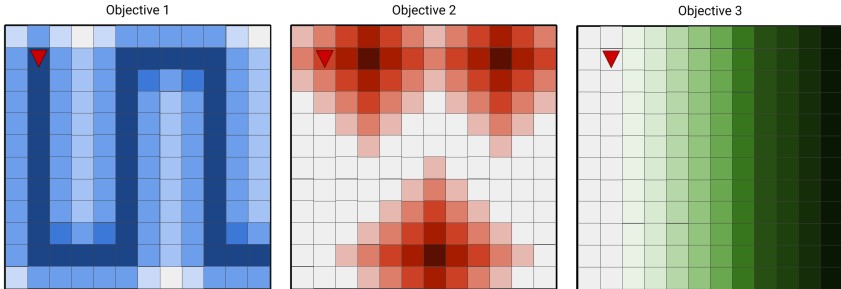

Figure 2: A visualization of the navigation environment. There are three objectives: 1) staying on the road, 2) avoiding hazards, and 3) moving to the right. Darker cells correspond to higher rewards in objectives one and three and lower rewards in objective two. All rewards are scaled between zero and one.

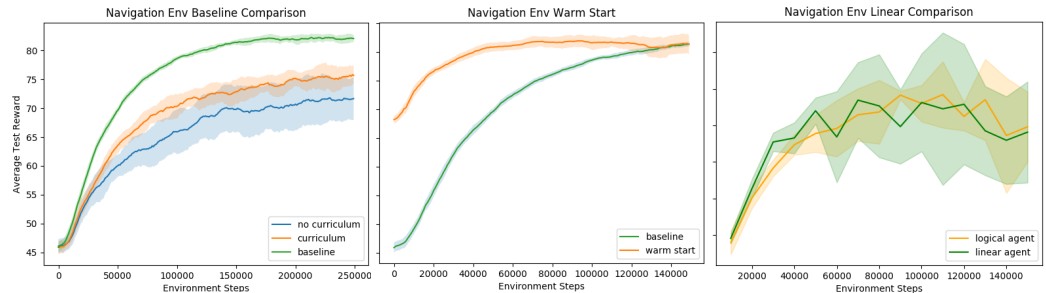

Figure 3: The left graph plots the average score for 100 never-before-seen test specifications over environment steps. The middle graph shows agents trained on a single specification initialized with parameters from the curriculum agent at 100,000 timesteps compared with the baselines for 100 averaged test specifications. The right graph compares the performance of our agent with the performance of a multi-objective agent trained with linear weight vectors.

transitions have a 10% chance of being random. The agent's initial state is indicated in Figure 2, one square down and to the right from the top left corner.

## 4.1 ZERO-SHOT GENERALIZATION RESULTS

Our initial experiments compare our agent to baseline agents trained on a single policy. For these experiments, we use the navigation environment defined previously with three objectives: stay on the road, avoid hazards, and move right. Note that the opposite of each of these objectives are also included in possible behavior specifications due to the semantics of our language that enable minimization. We define a set of 50,000 specifications to sample from during training, with which the agent learns to generalize to novel specifications. These specifications are randomly generated according to number of atomic statements, logical connectives, hard vs. soft constraints, and value of constraints. We randomly sample test specifications from these generated specifications. The left plot in Figure 3 shows the average episodic reward for 100 never-before-seen test specifications throughout training for our agent with and without curriculum learning. We compare the results of these two agents with 100 baseline DQN agents trained on each of these 100 behaviors. The error bars in Figure 3 show one standard deviation in average reward for multiple agent initializations.

Our results indicate that learning curricula with respect to specification length improve the speed and performance of learning and decrease the variance of learned policies. Our agents are able to learn near baseline policies for never-before-seen specifications with the same amount of experience and network updates that it takes baseline DQN agents to learn individual policies.

Figure 4 visualizes resulting reward functions and policies learned by our agent for three never-before-seen behavior specifications. The policies generated by our agent demonstrate an understanding of the specified behavior. For example, the specification shown in the left grid in Figure 4,

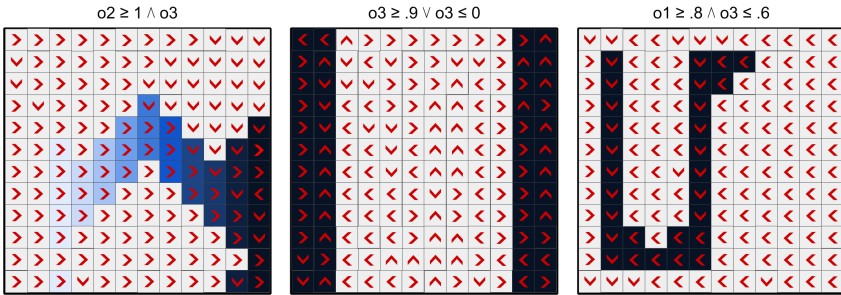

Figure 4: Zero-shot reward functions and policies for three never-before-seen test specifications. Darker cells indicate higher reward for entering that state. The left specifications instructs the agent to avoid hazards while moving to the right. The middle specification instructs the agent to move to the right two **or** left two columns. The right specification instructs the agent to stay on the road, while **avoiding** the four rightmost columns. See text for additional details.

$o2 \geq 1 \wedge o3$, can be interpreted as: avoid being within 4 squares of the marked hazards while maximizing the proximity to the rightmost column. The derived policy, focuses on arriving at the rightmost column of the grid where it will receive the greatest reward while moving away from the hazards when appropriate. The middle grid's specification, $o3 \geq .9 \vee o3 \leq 0$, instructs the agent to move to the right two or the left two columns. Note that the policy correctly sends the agent to the appropriate columns near the ends of the grid, but the center of the grid contains a few loops and other artifacts of conflicting policies. During execution, the agent eventually reaches one of the specified goals due to the stochasticity of the environment. The right grid shows the actual reward function and the generated policy for the specification $o1 \geq .8 \wedge o3 \leq .6$. This behavior requires the agent to stay near the road while avoiding the four rightmost grid columns. Notice that the agent's policy focuses on remaining on the road when to the left of the 4 rightmost columns, otherwise the agent is only concerned with avoiding those columns.

## 4.2 WARM-START TRAINING RESULTS

While the results in Figure 3 show impressive generalization to never-before-seen tasks, performance is not quite optimal. This begs the question: can the parameters learned when training on a variety of tasks be fine-tuned on a single task? Here, we test this by training a single policy agent initialized with parameters taken from one of our curriculum agent at 100,000 timesteps. We compare to the baseline DQN agent (which is always trained on a single task). The middle plot in Figure 3 shows this warm start agent compared to the baseline agents. The graph demonstrates that our agent trained to generalize over specifications has the ability to specialize to individual specifications much faster than a baseline agent with traditional, randomly initialized parameters.

## 4.3 LINEAR AGENT COMPARISON

Our agent learns a the non-linear scalarization function defined by the semantics of the propositional logic defined in Section 3.1. Most of the specifications that can be expressed in this language are not easily represented by linear weights. However, specifications that only use soft maximization constraints and logical *and* connectives can be expressed as weight vectors that contain only ones and zeros. For example, $o1 \rightarrow (1\ 0\ 0)$ and $o2 \wedge o3 \rightarrow (0\ 1\ 1)$. We use this method to compare our agent trained on logical specifications to agents from recent related work trained on linear weights. We compare the performance of each of these agents throughout training on the seven possible combinations of objectives according to the method just described. The linear agent uses the same parameters, architecture, and training algorithm as the logical agent but replaces sampled logical specifications with sampled weight vectors. We sample weight vectors from a Dirichlet distribution ($\alpha = 1$) following the method of Abels et al. (2018) who also utilize UVFAs for training their multi-objective agent. The right plot in Figure 3 shows the results of this experiment. Our linear agent was able to learn to satisfy the linear objectives with a speed and level of performance that mirrors the linear agent. These results indicate that our agent does not lose performance on traditional linearly weighted objectives while learning to satisfy more complex non-linear combinations of objectives.

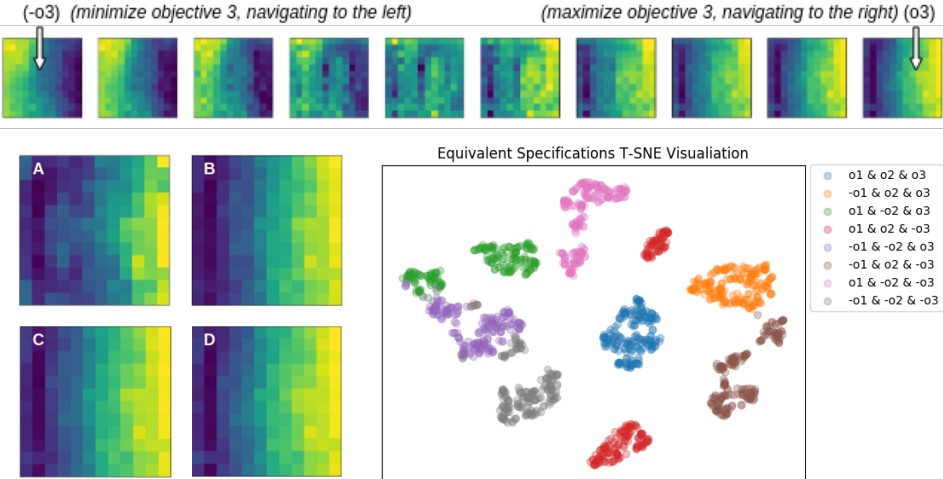

Figure 5: This figure provides several visualizations of encoded specifications. The top sequence of images are heat maps of the predicted state value from our Q-Network. The images interpolate between $-o3$ and $o3$ in specification encoding space showing that our agent learns smooth policy transitions between specifications. The bottom right image is a T-SNE visualization of 1,600 specification encodings. The encodings are organized into 8 buckets of 200 semantically equivalent specifications. The bottom left image shows the predicted state values for four semantically equivalent specifications: A) $-o3$  B) $(-o3)$  C) $((-o3))$  D) $((-o3 \vee -o3))$

## 4.4 SPECIFICATION ENCODING VISUALIZATIONS

Although our agent demonstrates impressive performance across a large number of specifications, we would like to be assured that the agent is actually learning the semantics of logical specifications and correctly estimating Q-values for specifications across states. The results of Figure 4 begin to demonstrate this by visualizing the learned policy for several logical specifications. We further demonstrate our agent's ability to learn semantics and implement policies across states and specifications through a number of experiments, the results of which are found in Figure 5. The top sequence of images in the figure contains heat maps of our agent's predicted values over environment states. The heat maps interpolate between the specifications $-o3$ and $o3$ from left to right by moving between the encodings of each in specification encoding space. Value clearly shifts from left to right as expected. Interestingly, traces of the first objective can be seen during the transition when the agent in not preferring left nor right.

The bottom right graph in the figure plots a T-SNE visualization of 1,600 encodings of logical specifications. These specifications are organized into 8 buckets of 200 semantically equivalent specifications. For example, the specifications $o1 \wedge o2 \wedge o3$ would be found in the same bucket as $o1 \wedge (o3 \wedge o2)$ or $(o1 \wedge o1) \wedge (o2 \wedge o3)$. We generate 200 unique specifications using the method described in Section 4.1 for each of the specifications listed in Figure 5's T-SNE visualization. The plot indicates that our agent is indeed consistently learning the semantics of various language specifications.

Finally, the bottom left grid in Figure 5 shows the state value heat maps for four semantically equivalent specifications $-o3$, $(-o3)$, $((-o3))$, and $((-o3 \vee -o3))$ labeled as A, B, C, and D respectively. The heat maps show that our agent finds similar state values (and thus policies) for semantically equivalent logcial specifications in addition to placing them close together in specification encoding space.

## 4.5 NAVIGATION ENVIRONMENT VARIATIONS

To test our agent on a variety of environments with various numbers of objectives we designed modified navigation environments. We scaled the environment to 5x5 and 20x20 grids as outlined in Figure 6 along with the 12x12 version described previously. We train on these modified navigation

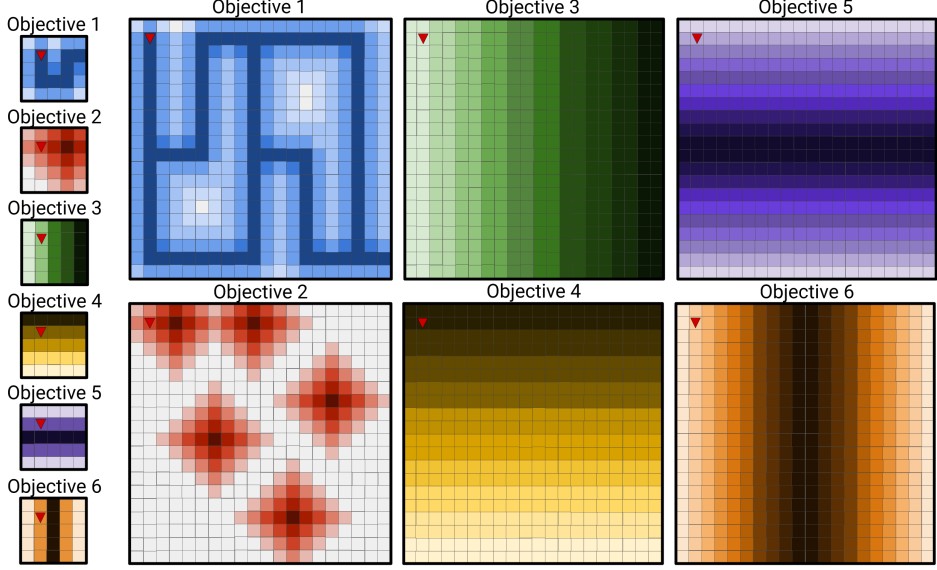

Figure 6: Visualizations of the small and large environments for all six objectives. Again, darker cells correspond to higher rewards for all cases but objective two. The medium environment, depicted in Figure 2, is also used with the additional objectives four, five, and six that follow the pattern shown here.

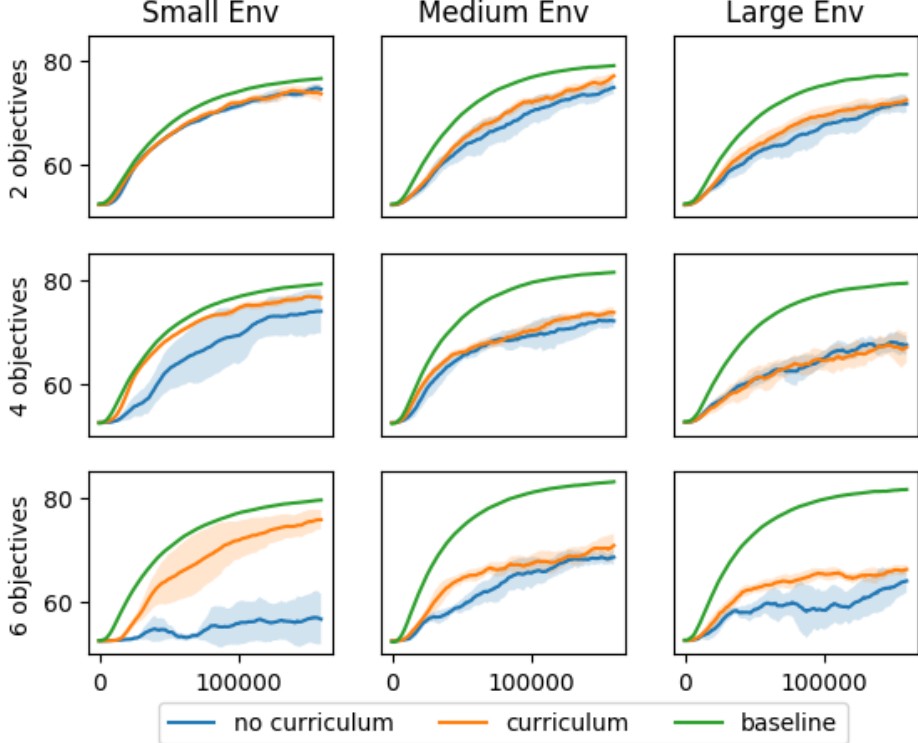

Figure 7: Agent performance compared to baselines for 3 different sized environments with 2, 4, and 6 objectives. The small, medium, and large environments are 5x5, 12x12, and 20x20 respectively. The medium environment is identical to the one described in the previous section, and the others are variations of it. The objectives are added in the following order: 1) stay on the road, 2) avoid hazards, 3) move right, 4) move up, 5) move towards the horizontal center, and 6) move towards the vertical center.

environments with two, four, and six objectives. The two objective environment includes staying on the road and avoiding hazards. The four objective environment adds moving to the right and moving up. The six objective environment adds moving towards the center row and moving towards the center column. With the latter four objectives, behavior specifications can include moving to arbitrary locations in the grid.

Figure 7 shows plots for the nine resulting combinations of environment size and objective count. In these experiments we use predefined sets of 40,000, 60,000, and 80,000 specifications with an 80:20 training-testing split for the two, four, and six objective versions respectively. We randomly sampled 20% of the specifications to form test sets and show that our agent again generalizes to never-before-seen behavior specifications over various environment sizes and objective counts. We found that increased environment size and objective count increased the generalization difficulty, as indicated by the increase in disparity between our agent and baseline performance, but our agent still manages to generalize in these more difficult environments. It is interesting to note that with increased number of objectives, the difference between non-curriculum and curriculum agents becomes more apparent.

## 5 Conclusions and Future Work

An ideal decision making agent has the ability to adjust behavior according to the needs and preferences of a user. Preferably, this would be done without retraining. Rather than learning separate policies for each desired behavior, we have shown that information about state, transitions, and the interactions between objectives can be captured and shared in a single model.

This work can be framed in many ways: as a non-linear generalization of MORL, or as a generalization of UVFA where value functions generalize across complex tasks specifications, instead of a single goal state. Either way, our results demonstrate that deep RL agents can successfully learn about, and generalize across, complex task specifications encoded in complex languages. We are particularly excited about our zero-shot results: our agents generalize to never-before-seen task specifications that are complex and nuanced, and are able to perform almost optimally with no task-specific training. If we allow the agent to train on new tasks, our warm-start experiments suggest that the parameters we have found serve as an excellent initialization that enables an agent to rapidly achieve optimal performance on a new task.

We have also shown preliminary evidence that when trying to summarize and utilize complex specifications, some sort of curriculum-based learning is likely to be necessary. By increasingly complexifying the task specifications, we were generally able to achieve better, more stable, and lower-variance behavior.

In this work, our MORL agent demonstrates that it is possible to generalize over language defined specifications and behave well when given completely novel behavior specifications. This is possible in part through learning semantic representations of language expressions and then generalizing over those representations, an idea that can be generalized to other languages.

This naturally begs the question: could we encode task specifications using natural language, instead of a logical language? Previous attempts have been made to combine natural and formal language with RL. Natural language has been used for advice giving (Kuhlmann et al., 2004), reward shaping (Goyal et al., 2019), and defining reward functions (Fu et al., 2019) in RL tasks. Quantitative semantics of temporal logics have been used to define reward functions (Aksaray et al., 2016; Li et al., 2017) and ensure safe exploration (Li & Belta, 2019). More complex languages like these would allow for more complex behaviors. Finally, in this work, we use a simple language based on propositional logic to specify desired behavior, but in future work, MORL scalarization functions may include temporal and other formal logics. Temporal logics would allow for complex sequences of commands with respect to multiple objectives, becoming much more applicable to real world scenarios. Natural languages come with additional difficulties due their lack of concrete semantics. However, improvements in natural language representation may also enable the use of natural language to specify desired behavior in MORL.

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
