# OpenReview forum: "Using Logical Specifications of Objectives in Multi-Objective Reinforcement Learning"
_ICLR.cc/2020/Conference — Reject_

### Official Review · AnonReviewer2 · 2019-10-22
**Official Blind Review #2**

**Rating:** 6

**Review:**

This paper proposes using logical specifications to facilitate Q-learning in multi-objective reinforcement learning (MORL). Empirically the proposed method can generalize to unseen reward specifications with performance competitive to agents being fully trained in the new environment. The proposed setting employs a more expressive objectives space induced by propositional logic. The proposed method uses a recurrent encoder to embed specifications into vectors and uses them to parametrize the Q-function.

Overall, I weakly recommend accepting the submission for the following reasons: (+) it proposes using propositional logic to specify reward functions, which broadens the objective space in an interpretable way, (+) the learned objective embedding demonstrates the ability to generalize to unseen environments(rewards). However, there is still room for improvement. I will raise my score if the following problems are addressed: (-) Needs more diverse experiments (instead of grid worlds) to support the paper (-) It might be hard to express objectives using logic formulas in real-world applications.

More specifically, the problems of the paper are,

(-) The scalability issue. The most complicated problem in the experiments has a 20x20 state space, which is pretty small for a typical RL problem. I wonder whether the model is still generalizable for larger problems. Even in the case of 20x20 grids, we can notice the gap between the baseline and the proposed method.

(-) The assumption. This paper assumes the logical specifications are given by human. However, we usually don't know how to describe the true objective with logic formulas. Sometimes, finding the specification(reward) itself is as difficult as finding a good policy. Is it possible that we can relax this assumption?

Minor comments:
Figure 6: The word "baseline" here is misleading. A better choice would be "upper-bound"?

**Experience Assessment:**

I have read many papers in this area.

**Review Assessment: Checking Correctness Of Derivations And Theory:**

N/A

**Review Assessment: Checking Correctness Of Experiments:**

I carefully checked the experiments.

**Review Assessment: Thoroughness In Paper Reading:**

I read the paper thoroughly.

---

> ### Author Response · Authors · 2019-11-15
> **Paper Revision: New Sections 4.3 and 4.4**
>
> Thank you for your positive feedback and your criticisms. As you pointed out, our agent was better able to generalize in some environments over others. Since our original submission, we were able to continue several of these tests and saw that our agent continued to approach the upper-bound performance. An updated graph is included in the revised version of our paper. It is clear that more complicated environments require longer training time to reach upper-bound performance, but it is also clear that some of our agents were did not yet train to convergence. We expect that our agent would continue to approach the desired performance if we trained them for even longer.
>
> However, we would like to reiterate that we did not expect our agent to perform as well as agents trained on a single policy. The fact that our agent generalizes to novel specifications and approaches the performance of agents trained specifically for those specifications is notable in itself. As we demonstrated in Figure 3, our agent also serves as an excellent warm start for specifications that it was never trained on.
>
> Regarding your suggestion to relax the assumption that logical specifications are human given, we believe this to be an interesting problem for future research but outside of the scope of this paper. Like you said, we assume that the reason for doing multi-objective RL is that the user does not know the desired behavior of the agent prior to training. After training, a user can then specify a logical statement to sclarize the multi-objective reward vector. Our purpose was to reduce the strain of specifying how multiple objectives should be scalarized. It is easier to specify desired behavior with logical specifications than with linear weight vectors, and our research shows that our agents are performs well for a large test set of never-before-seen specifications.
>
> We encourage you to take note of our additional Sections 4.3 and 4.4  as described in our comment to all reviewers as we believe that they are further evidence of the success of our method.

---

### Official Review · AnonReviewer1 · 2019-10-23
**Official Blind Review #1**

**Rating:** 3

**Review:**

The authors propose to tackle the problem of multi-objective reinforcement learning (MORL) by considering a logical function as reward signal. In their proposed solution, the logical function is also encoded and concatenated with the state. They argue, via simulation and toy examples, that the proposed model is able to generalize to logical formulas of the multidimensional rewards that has not observed during training.

The combination of logic and machine learning is certainly an interesting direction. However, the current contribution is rather limited in terms of methodology and shallow in terms of experimental evaluation. As a result, I do not support acceptance. More specifically:

1. The only methodological novelty of proposed contribution is the idea of encoding the multi objective reward as a logical function. As a result, the experimental evaluation should be much more thorough.

2. The general claims made by the authors are not really supported by the experimental evaluation. In particular, many details of their experimental setup are missing (e.g., the experiments use 50,000 specifications, however, no stats about these specifications are given), the experiments are mainly about performance rather than exploring the encodings, and thus it is difficult to judge whether the proposed solution is actually meaningfully encoding the logic and achieve generalization.

3. In the experiments, the objectives stay on the road, avoid hazards and move right are not clearly specify mathematically. What is the range of o1, o2, o3? o1, o2, o3 are sometimes used in inequalities, sometimes they are used as Boolean variables. The authors should more clearly explain this.

Minor comments: "agent for use with" -> "agent to use" (Just a couple of paragraphs before Background)

**Experience Assessment:**

I have read many papers in this area.

**Review Assessment: Checking Correctness Of Derivations And Theory:**

I assessed the sensibility of the derivations and theory.

**Review Assessment: Checking Correctness Of Experiments:**

I assessed the sensibility of the experiments.

**Review Assessment: Thoroughness In Paper Reading:**

I read the paper at least twice and used my best judgement in assessing the paper.

---

> ### Author Response · Authors · 2019-11-15
> **Paper Revision: New Sections 4.3 and 4.4**
>
> We thank you for your comments. Regarding your first point, we are currently preparing additional experiments to further solidify our experimental evaluation. Unfortunately there are not many multi-objective RL frameworks, and other RL frameworks that rely on visual observations make extracting multiple objectives difficult. For this reason we had to design our own experiments as other multi-objective RL research has done recently.
>
> First, regarding your concern about meaningful encodings, we have run additional experiments, and we are very happy with the results visualized in Figure 5. We set out to more clearly show that our agent is meaningfully encoding the logic of each specification by demonstrating that our encoder does learn the semantics of specifications and correctly uses this to predict meaningful state values. We show this by plotting the T-sne visualizations of sets of semantically equivalent specifications. We also visualize heat maps of state values interpolated between two specifications. We ask that you consider the significance of these visualizations as they clearly indicate that our method is able to learn and represent the semantics of logical statements in a way that can then be used to generate useful policies.
>
> We urge you to consider the significance of the unique, interpretable, and non-linear scalarization function that is implicit in the quantitative semantics of logical specifications. We are adding additional results comparing the performance of our agent with traditional linear multi-objective agents on never-before-seen linear specifications. These results indicate that we are able to learn a much wider range of both linear and non-linear policies compared to traditional linear multi-objective agents. Thus our method is not only a different representation of the same objectives, but allows for much richer expression of desired objectives.
>
> We appreciate your second comment about how we left out the exact nature of the specifications we used for training and testing. We are including this information in the revised version of paper. Essentially, we use a script to randomly generate unique and specifications. We use the same script to generate both training and testing specifications. Examples are given in the revised version of the paper.
>
> The answers to most of the questions in your third comment are found in section 3.1 of the paper. In our experiments, the values of each objective stay in the range 0 to 1 inclusive. o1, o2, and o3 are always used as variables, but the semantics of our language specify both soft constraints, where the objective will be evaluated with a value between 0 and 1, and hard constraints, where the value of an objective will be evaluated at either 0 or 1. The quantitative semantics of the language specify our various objectives are to be scalarized. The final scalar value produced by a reward vector and logic specification is then maximized by the agent. For further details on the exact values of each state, a reader can reference our included code link. Thank you for your review.

---

### Official Review · AnonReviewer3 · 2019-10-28
**Official Blind Review #3**

**Rating:** 3

**Review:**

Thank the authors for the response. The major novelty of this paper is encoding the objective as a logical expression and the experiment part is limited. I will keep my score.
----------------------------------------
Summary
This paper presents a new approach for MORL (Multi-Objective Reinforcement Learning), which handles multi-objective as a pre-defined logical language, and estimates Q-function by using UVFA. This approach can handle the case when the final objective is not a linear combination of the individual objectives. It feeds the specification into a GRU (Gated Recurrent Units) as an encoder, concatenate the encoding with the state, and then send it to the Q-function, estimated by a UVFA. I am kind of on the borderline, but still lean to reject this paper. I am happy to change my score based on the reviews from other reviewers.
Strengths
- The idea of modeling multi-objective as a logical language is novel. By describing the final objective in a logical language, many more cases that are not linear can be covered.
Weaknesses
- Lack of baselines / experiments. It seems the only baseline in this paper is agents trained on a single policy, i.e., no baseline from previous works. It is possible to compare the performance with previous works when the final objective is linear to each individual objective.
- Only tested on one simple scenario. More scenarios can be included to justify the effectiveness of the proposed approach (e.g., Deep Sea Treasure, SuperMario, etc.).
Possible Improvements
As mentioned before, more baselines and scenarios can be included.

**Experience Assessment:**

I have published one or two papers in this area.

**Review Assessment: Checking Correctness Of Derivations And Theory:**

I assessed the sensibility of the derivations and theory.

**Review Assessment: Checking Correctness Of Experiments:**

I assessed the sensibility of the experiments.

**Review Assessment: Thoroughness In Paper Reading:**

I read the paper at least twice and used my best judgement in assessing the paper.

---

> ### Author Response · Authors · 2019-11-15
> **Paper Revision: New Sections 4.3 and 4.4**
>
> Thank you for your review. We originally did not compare our method with previous work in multi-objective RL because our non linear specifications are not learnable by other linear agents. However, in the revised version of our paper we compare our agent’s performance on several linear logical specifications with the performance of a linear multi-objective RL agent. We model this linear agent after recent work published at ICML 2019 that we reference in our paper. We were very pleased that our results indicate that our agent generalizes to linear specifications with performance nearly identical to strictly linear agents trained with weight vectors. We discuss this in the new Section 4.3.
>
> Unfortunately not many learning environments exist for training multi-objective RL agents. Popular environments that rely on complex observations such as game screenshots so not provide enough information about the game state to extract multiple objective reward signals. However, we are preparing more complex multi-objective environments that we will be running tests on in the near future.
>
> We encourage you to take note of our additional Section 4.4  as described in our comment to all reviewers as we believe that they are further evidence of the success of our method.

---

### Author Response · Authors · 2019-11-15
**Paper Revision: New Sections 4.3 and 4.4**

We thank the reviewers for their excellent feedback and provide this revised version of our paper. We tried to address some of the points that needed further clarification such as language semantics and training procedure. We updated Figure 6 (now Figure 7) with additional tests that extend the length of training in each environment and add error bars from several different random initializations of agents.

Additionally, we added a graph to Figure 3 that compares the performance of our agent with the performance of a multi-objective agents trained on linear weightings of objectives similar to recent related work. Discussion of this experiment can be found in the new Section 4.3. We compared these two agents on several specifications that are easily represented as both logical statements and linear weightings (for example: “o1 & o3” and “[1,0,1]”). Our results indicate that our agent learns linear specifications just as well as agents from related work.

Finally, we were very pleased with our attempts to illustrate that our method is meaningfully encoding the logic within specifications motivated by some of your comments. The new Figure 5 and Section 4.4 demonstrate that our encoder does learn the semantics of specifications and correctly uses this to predict meaningful state values. We show this by plotting the T-SNE visualizations of sets of semantically equivalent specifications. We also visualize heat maps of state values interpolated between two specifications. We ask that you consider the significance of these visualizations as they clearly indicate that our method is able to learn and represent the semantics of logical statements in a way that can then be used to generate useful policies.

We thank you again for your input and suggestions.

---

### Decision · Program_Chairs · 2019-12-19

**Decision:**

Reject

**Comment:**

The reviewers generally agreed that the technical novelty of the work was limited, and the experimental evaluation was insufficient to make up for this, evaluating the method only on relatively simple toy tasks. As much, I do not think that the paper is ready for publication at this time.